# Green Tattoo Pre-Operative Renal Embolization for Robotic-Assisted and Laparoscopic Partial Nephrectomy: A Practical Proof of a New Technique

**DOI:** 10.3390/jcm11226816

**Published:** 2022-11-18

**Authors:** Eliodoro Faiella, Alessandro Calabrese, Domiziana Santucci, Riccardo Corti, Nicola Cionfoli, Claudio Pusceddu, Carlo de Felice, Giorgio Bozzini, Federica Mazzoleni, Rosa Maria Muraca, Lorenzo Paolo Moramarco, Massimo Venturini, Pietro Quaretti

**Affiliations:** 1Department of Radiology, Sant’Anna Hospital, Via Ravona, San Fermo della Battaglia, 22042 Como, Italy; 2Department of Radiological Sciences, Oncology and Pathology, Umberto I Hospital, Sapienza University of Rome, Viale del Policlinico 105, 00161 Rome, Italy; 3Unit of Computer Systems and Bioinformatics, Department of Engineering, Campus Bio-Medico University, Via Alvaro del Portillo 21, 00128 Rome, Italy; 4Unit of Interventional Radiology, IRCCS Policlinico San Matteo Foundation, 27100 Pavia, Italy; 5Regional Referral Center for Oncologic Disease, Department of Oncological and Interventional Radiology, Businco Hospital, A.O. Brotzu, 09100 Cagliari, Italy; 6Department of Diagnostic and Interventional Radiology, Ospedale di Circolo e Fondazione Macchi, University of Insubria, 21100 Varese, Italy

**Keywords:** renal tumors, indocyanine, embolization, green-tattoo technique

## Abstract

(1) Background: Our aim is to describe a new mixed indocyanine-non-adhesive liquid embolic agent (Onyx-18) pre-operative renal embolization technique for assisted-robotic and laparoscopic partial nephrectomy with near-infra-red fluorescence imaging. (2) Methods: Thirteen patients with biopsy-proven renal tumors underwent pre-operative mixed indocyanine–ethylene vinyl alcohol (EVOH) embolization (Green-embo) between June 2021 and August 2022. All pre-operative embolizations were performed with a super selective stop-flow technique using a balloon microcatheter to deliver an indocyanine-EVOH mixture into tertiary order arterial branch feeders and the intra-lesional vascular supply. Efficacy (evaluated as complete embolization, correct tumor mapping on infra-red fluorescence imaging and clamp-off surgery) and safety (evaluated as complication rate and functional outcomes) were primary goals. Clinical and pathological data were also collected. (3) Results: Two male and eleven female patients (mean age 72 years) received pre-operative Green-embo. The median tumor size was 29 mm (range 15–50 mm). Histopathology identified renal cell carcinoma (RCC) in 9 of the 13 (69%) patients, oncocytoma in 3 of the 13 (23%) patients and sarcomatoid RCC in 1 of the 13 (8%) patients. Lesions were equally distributed between polar, meso-renal, endo- and exophytic locations. Complete embolization was achieved in all the procedures. A correct green mapping was identified during all infra-red fluorescence imaging. All patients were discharged on the second day after the surgery. The median blood loss was 145 cc (10–300 cc). No significant differences were observed in serum creatinine levels before and after the embolization procedures. (4) Conclusions: The Green-tattoo technique based on a mixed indocyanine-non-adhesive liquid embolic agent (Onyx-18) is a safe and effective pre-operative embolization technique. The main advantages are the excellent lesion mapping for fluorescence imaging, reduction in surgical time, and definitive, complete and immediate tumor devascularization based on the deep Onyx-18 penetration, leading to a very low intra-operative blood loss.

## 1. Introduction

In the past 20 years, partial nephrectomy has replaced radical nephrectomy in the surgical treatment of small renal masses (less than 4 cm in size) [1]. In small renal masses, partial nephrectomy allows oncologic outcomes similar to those of radical nephrectomy while ensuring good long-term renal function and avoidance of chronic kidney disease, without compromising overall survival [1,2,3]. To ensure minimal invasiveness, the current procedural standard for partial nephrectomy is the laparoscopic surgery approach. Despite the benefits of partial nephrectomy, this technique is a more complex procedure, as it is burdened by longer warm ischemia times, a higher rate of urinary system damage, and a higher rate of complications than radical nephrectomy, especially in the case of complex renal tumors, increasing the risk of chronic kidney disease [4].

The introduction of robot assistance has helped with the difficulties of laparoscopic partial nephrectomy. Robot-assisted partial nephrectomy has maintained minimal invasiveness, achieved a reduction in peri- and post-operative complications, and warm ischemia times with less estimated blood loss and shorter hospital stays, further improving long-term renal function [5]. Resection of entirely endophytic renal masses is more challenging because of the complexity of the tumors, making intraoperative tumor identification and recognition of the mass extension more difficult [6].

Several studies demonstrated improved survival and reduced blood loss in patients who underwent renal artery embolization before partial nephrectomy surgery, with further reduced rates of post-operative complications and consequently shorter hospital stays [7,8,9]. Almgård et al. popularized selective embolization of the renal artery during the 1970s, producing necrosis of the tumor mass and a reduction in vascularity, leading to a reduction of tumor size and peri-operative bleedings [10]. 

A considerable variety of embolizing agents can be used in renal artery embolization. Renal vessels are considered arteries without significant collaterals, and renal masses are often hypervascular tumors with abundant neoangiogenesis and recruitment of extra-renal vasculature. Therefore, embolizing agents that result in permanent occlusion of small vessels, such as PVA, ethanol, or microspheres are preferred [11]. More recently, liquid embolizing agents, such as Onyx (Covidien, Plymouth, MN, USA), Precipitating hydrophobic injectable liquid (PHIL), and Squid™/Squid Peri™ (Emboflu, Epalinges, Switzerland) have been reported as being used in the pre-operative treatment of renal masses [12,13,14]. 

Near-infrared fluorescence imaging (NIFI) with indocyanine green (ICG; IC-Green; Akorn, Lake Forest, IL, USA) is used to better visualize anatomical structures during surgery by differentiating normally-perfused renal parenchyma (which appears green) from the renal mass, even in completely endophytic lesions. NIFI in robot-assisted partial nephrectomy can be performed safely, with better short-term results for renal function [15]. Few studies exist regarding the NIFI approach with ICG in renal artery embolization before robotic-assisted and laparoscopic partial nephrectomy [16,17].

The aim of this study is to describe a novel pre-operative renal embolization technique with a mixed indocyanine-non-adhesive fluid agent (OnyxTM 18 LES) for robotic and laparoscopic-assisted partial nephrectomy with NIFI, assessing the safety and efficacy of the ultimate and instantaneous penetration of this embolizing fluid agent.

## 2. Materials and Methods

### 2.1. Study Design and Population

Thirteen patients, candidates for robotic-assisted and laparoscopic partial nephrectomy, received pre-operative mixed indocyanine–ethylene vinyl alcohol (EVOH) (Green-embo) embolization between June 2021 and August 2022. All the patients enrolled in the study presented these inclusion criteria: (a) pre-operative contrast-enhanced computer tomography (CECT) and/or contrast-enhanced MRI; (b) biopsy-proven renal mass; (c) laparoscopic or robot-assisted partial nephrectomy according to current guidelines.

This is a retrospective observational study; only existing information collected from human participants was used, and there are no identifiers linking individuals to data/samples. Institutional Review Board approval has been obtained. All methods and procedures conform to the ethical standards of the institution and the research committee, in accordance with the 2013 Declaration of Helsinki. The clinical and pathological data of patients included in the study are shown in Table 1.

### 2.2. Embolization Technique

Before the embolization procedure, a mixture consisting of indocyanine (powder 25 mg, diagnostic green) and OnyxTM 18 LES (Medtronic, Minneapolis, MN, USA) is prepared. 

The operator has one vial of DMSO, one vial of Onyx-18 and one vial of indocyanine in powder available.

The first step is to septically withdraw 0.7 mL from the DMSO vial and inject them into the container of indocyanine powder, mixing.

Next, 0.5 mL of the new compound consisting of DSMO and indocyanine is taken and administered into the vial containing Onyx-18.

The new mixture, consisting of 0.5 mL of DSMO-indocyanine and a vial of ONYX-18, is obtained after being mixed automatically for about 20 min and will be used to carry out the pre-operative embolization.

The residual volume of DSMO and possibly another vial of DSMO, if necessary, will be used to fill the dead volume of the microcatheter before the administration of the embolizing mixture. The procedure was performed in the angiographic suite by two experts IR. After the right inguinal puncture and placement of a 6Fr femoral sheath, a 6 Fr guiding catheter (Chaperon 6 Fr Microvention, Envoy 6 Fr Codman, or Mach 1 6 Fr Boston Scientific) was placed in the renal artery and an angiographic assessment of the kidney vascular supply was performed. Then, super selective catheterization of the lesion feeders was performed using a balloon microcatheter (Scepter XC, Microvention, CA, USA) and a guide wire (Transend standard 0.014 in, Stryker, Kalamazoo, MI, USA). After the inflation of the microballoon, a control run was performed from the microcatheter to determine the precise position of the distal tip and simulate embolic injection.

Embolization was then performed using the DSMO–indocyanine–OnyxTM 18 LES mixture (Medtronic) under slow infusion, using a blank roadmap visualization to achieve distal penetration as anatomically possible until complete flow stasis was achieved within each feeding vessel. 

The administration of the embolizing agent with indocyanine is performed in stop-flow mode by balloon microcatheter so that the administration is modulated based on the force impressed by the operator and the risk of non-target embolization or reflux is minimized. In addition, the drop in pressure gradient at the tip of the insufflated microcatheter allows a reacceleration of intralesional flow by improving the penetration of the indocyanine-embolizing fluid mixture into the lesion. In fact, the final effect is precisely that of a tattoo, hence the appellation of “Green Tattoo”.

At the end of the procedure, a control angiography was performed to assess the percentage of embolized tumor-feeding vessels. Successful embolization, considered as complete devascularization of the tumor, was finally evaluated by cone beam computer tomography (CBCT) or control contrast-enhanced ultrasound (CEUS).

Efficacy, evaluated as complete embolization, correct tumor mapping on infra-red fluorescence imaging and clamp-off surgery, and safety, determined as complication rate and functional outcomes, were the primary goals.

### 2.3. Statistical Analysis

Continuous variables were tested for normality using the Shapiro–Wilk test and presented as mean and range. Categorical variables were reported using counts and percentages. The Chi-square test or Fisher’s exact test was performed to assess the presence of statistically significant differences between categorical variables. A Pearson product-moment correlation was run to determine the relationship between serum creatinine levels before and after embolization. All data analyses were processed using SPSS (version 25.0; IBM, Armonk, NY, USA).

## 3. Results

Thirteen patients (2 males and 11 females, mean age 72 years) received pre-operative Green-embo. The median tumor size was 29 mm (range 15–50 mm). 

Lesions were equally distributed between polar, meso-renal, endo- and exophytic locations. 

Complete super selective embolization was achieved in all the procedures. No peri- or post-procedural complications were observed. The mean time of embolization was 46.5 min.

A correct green mapping was identified during all the infra-red fluorescence imaging. All patients were discharged on the second day after the surgery. R0 was achieved at surgery in all patients. Histopathology identified renal cell carcinoma (RCC) in 9 of the 13 (69%) patients, oncocytoma in 3 of 13 (23%) patients and sarcomatoid RCC in 1 of the 13 (8%) patients.

No significant differences were observed in serum creatinine levels before and after the embolization procedures.

The median blood loss was 145 cc (10–300 cc). Only one patient (1/13, 7%) had postoperative surgical complications (hematuria with anemia three days after the procedure, requiring hospitalization). Figure 1, Figure 2, Figure 3, Figure 4, Figure 5, Figure 6 and Figure 7 show three cases of patients who underwent the Green-tattoo technique.

## 4. Discussion

The current standard of practice in the surgical treatment of renal masses smaller than 4 cm consists of partial nephrectomy, which supplanted radical nephrectomy, with which it shares similar survival outcomes [1]. Laparoscopic partial nephrectomy has replaced open partial nephrectomy, compared with which it has similar outcomes on renal function, with shorter hospital stay times due to lower invasiveness; despite these benefits, it is a more complex technique than open partial nephrectomy, particularly in view of the higher rate of intra-operative complications, especially in larger tumors. Robot-assisted partial nephrectomy has numerous advantages over traditional laparoscopic nephrectomy in that it offers equivalent oncologic outcomes with reduced hospital due to less intraoperative blood loss and shorter warm ischemia time, leading to better renal function [5,18].

Gallucci et al. proposed super selective embolization of the tumor vessels before partial nephrectomy, to reduce the risk of bleeding, as the first step to off-clamp partial nephrectomy [18]. Although preoperative embolization of RCCs to reduce the tumor mass and facilitate surgery and reduction of peri-procedural bleeding was initially proposed by Almgård in 1973 [10], there is no present consensus or guidelines on its use in clinical practice. A meta-analysis by Shanmugasundaram et al. confirmed that renal embolization prior to surgical resection is a safe technique and effective at significantly reducing intra-operative blood loss in patients undergoing partial nephrectomy for RCC, particularly if the time interval from embolization to surgery is less than 48 h [19]. In addition, the rate of major complications is significantly lower in patients undergoing embolization followed by partial nephrectomy than in those undergoing partial nephrectomy alone [19].

Several intra-operative techniques have been introduced to facilitate tumor localization, differentiate tumors from healthy renal parenchyma, and evaluate resection margins. One of these techniques is intraoperative ultrasonography (US), which has long been used during conventional and robotic laparoscopic nephrectomy sessions, partly because of its wide availability. The weaknesses of in vivo intraoperative US include the need for the probe to be in contact with renal tissue during the procedure, making it less practical in real-time and slightly prolonging ischemia times to ensure proper assessment of resection margins [20].

Intraoperative NIFI is a relatively new development, with ICG being the most commonly used fluorescent dye in renal tumor surgery [21]. As healthy renal parenchyma accumulates ICG, injection of ICG can be used as a negative contrast agent to differentiate healthy renal parenchyma from renal masses such as RCC, which appears hypofluorescent at NIFI [21,22]. Furthermore, injection of ICG can confirm the presence of adequate ischemia (which appears hypofluorescent compared to perfused parenchyma) during selective clamping of the renal artery, thus reducing the proportion of renal parenchyma that is subjected to ischemia, and further reducing warm ischemia times [23]. ICG is not subjected to renal clearance, so it can be safely used in patients with reduced renal function and during the intra-operative period of renal ischemia.

One of the weaknesses of NIFI is represented by ICG-limited tissue penetration, which makes this technique less effective in endophytic tumors [24]. For this reason, Simone et al. tested a novel technique by selecting 10 patients with fully endophytic renal masses and administering an ICG-lipiodol mixture followed by robot-assisted partial nephrectomy [16]. This technique allowed rapid tumor identification during robotic laparoscopic partial nephrectomy, simplifying the procedure and allowing real-time control of resection margins without peri-operative complications and with negligible loss of renal function. Nardis et al. [17] evaluated the safety, efficacy, and clinical impact of a larger cohort of patients undergoing robotic-assisted partial nephrectomy preceded by administration of an ICG–lipiodol mixture. In addition, they reviewed the technique proposed by Simone et al. by performing intraprocedural cone-beam computed tomography to allow accurate assessment of vascular anatomy and evaluate the post-procedural deposition of the embolic material.

Super selective embolization of the renal artery with ICG and liquid embolization agents can offer numerous advantages in the practice of laparoscopic partial nephrectomy. First, super selective injection of ICG allows rapid and immediate identification of the tumor mass, even in cases of completely endophytic tumors, by tattooing the renal mass with indocyanine, making it hyperfluescent (“Green Tattoo”). This results in a lower average procedure time, with an average enuclueation time of about 10–15 min, versus an average operative time in partial nephrectomy with traditional clamping of 121.8 min. No difference in oncological outcomes was found between endophytic and exophytic lesions, since R0 was achieved in all lesions. This technique is therefore faster and more practical than intra-operative US, as it does not require the constant presence of the probe, which can interfere with resection instruments.

Administration of the embolizing agent allows the surgical procedure to be performed without renal pedicle clamping, thus causing no suffering to the renal parenchyma and allowing excellent renal function post-procedure. The high intra-lesional penetration is proved by the presence of the ICG-Onyx-18 mixture in the renal tumor vascular supply; the entity of the embolization performance is strictly connected with the dimension and the number of the renal tumor supply branches mapped on the pathology specimen as showed in Figure 2d, Figure 3e, Figure 5c and Figure 7d. In our study, no statistically significant differences were demonstrated between serum creatinine levels before and after the embolization procedure. In addition, the embolizing agent injected in stop-flow mode by balloon microcatheter, minimizing non-target embolization and promoting drug penetration, allowed excellent control of hemostasis; this resulted in a significant reduction in intra-procedural blood loss, with a median blood loss in our study of approximately 145 cc (range: 10–300 cc), which is slightly lower than the 250 cc reported by Simone et al. [16]. Furthermore, only one patient (1/13, 7%) had postoperative surgical complications.

Our study has the following limitations. It is a retrospective study, and only patients who underwent the mixed indocyanine-non-adhesive fluid agent renal embolization procedure for robotic partial nephrectomy and NIFI were included, so a direct comparison between the outcomes of our procedure and those of conventional and robotic laparoscopy with renal peduncle clamping was not possible. The small size of the study population and the short follow-up period do not allow evaluation of late oncologic outcomes and effects on renal function, nor do they allow comparison with established techniques. In addition, the costs of the procedure compared with conventional techniques have not been studied, although the fact that all patients were discharged on the second day after the procedure may allow for a reduction in overall expenditures and increased availability of hospital beds. Future studies with a larger population sample, multicenter, and prospective designs are needed for further evaluation and standardization of the technique and its diffusion in clinical practice.

## 5. Conclusions

The Green-tattoo technique based on a mixed indocyanine-non-adhesive liquid embolic agent (Onyx-18) is a safe and effective pre-operative embolization technique.

The technique is based on the deep penetration ability of a liquid embolic agent carrying ICG into the tumor vascular supply, resulting in excellent lesion devascularization and intra-operative tumor visualization and mapping with fluorescence imaging.

The main advantages are the excellent lesion mapping for fluorescence imaging, reduction in surgical time, and a definitive, complete and immediate tumor devascularization based on the deep Onyx-18 penetration by balloon microcatheter, leading to a very low intra-operative blood loss and rare post-procedural complications.

## Figures and Tables

**Figure 1 jcm-11-06816-f001:**
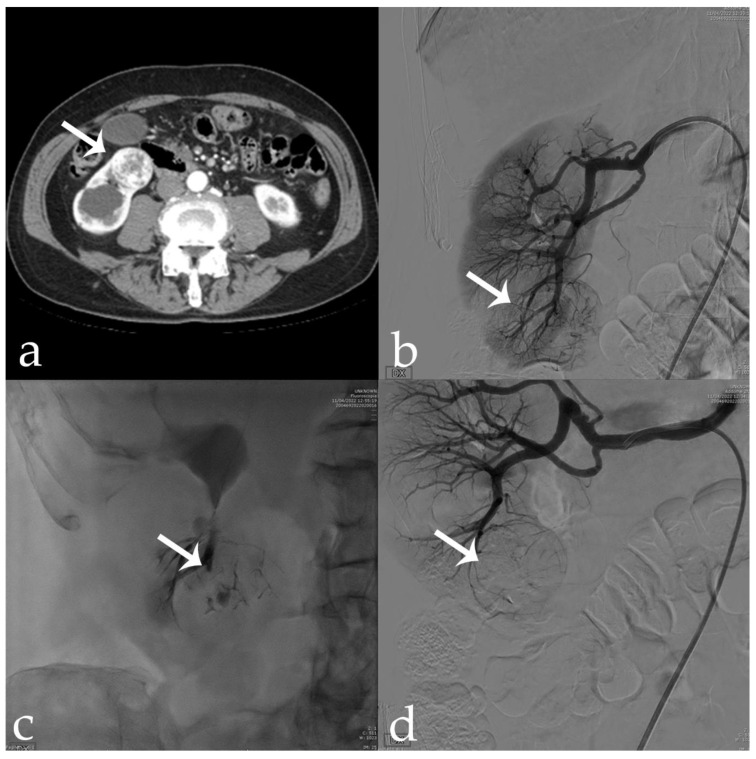
78 yr old male with exophytic RCC of the anterior aspect of the lower pole of the right kidney (46 × 35 mm) (arrows): (**a**) pre-operative CECT; (**b**,**c**) tumor vascular supply on DSA evaluation; (**d**) post-embolization control showing complete devascularization and deep penetration of the 18-Onyx and indocyanine mixture.

**Figure 2 jcm-11-06816-f002:**
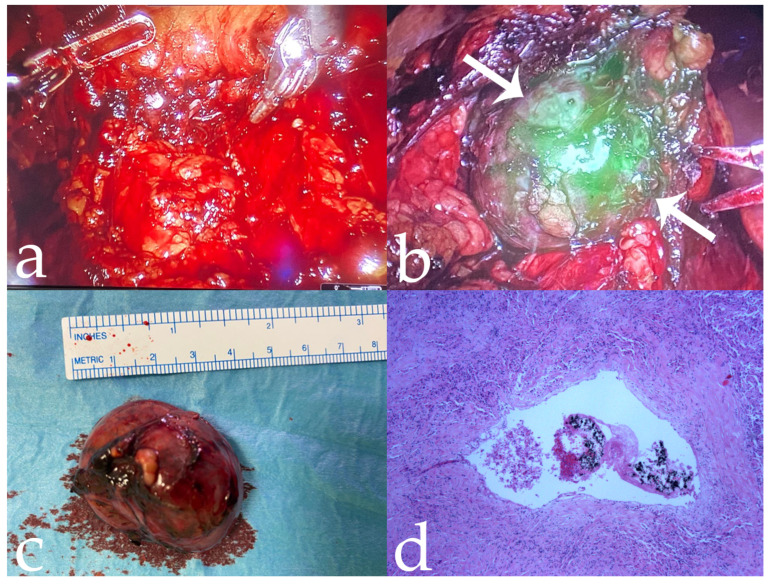
Same case as Figure 1. (**a**,**b**) Intra-operative tumor visualization before and after fluorescence imaging (arrows); (**c**) complete robotic-assisted tumor enucleation fluorescence imaging; (**d**) histological specimen confirming intralesional penetration of 18-Onyx (10×).

**Figure 3 jcm-11-06816-f003:**
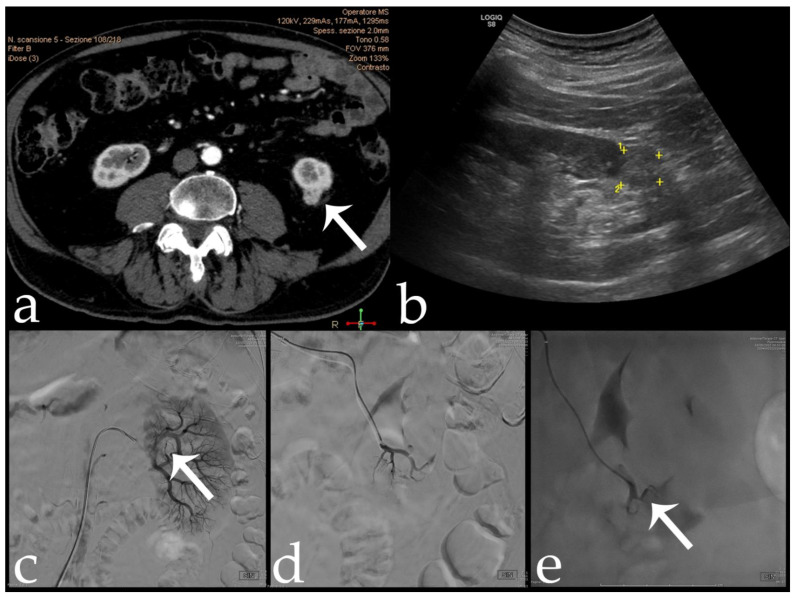
80 yr old male with intra-cortical RCC of the left kidney lower pole (15 × 15 mm) (arrows): (**a**,**b**) pre-operative CECT and US; (**c**) pre-embolization angiographic study; (**d**,**e**) post-embolization control after 18-Onyx and indocyanine mixture administration.

**Figure 4 jcm-11-06816-f004:**
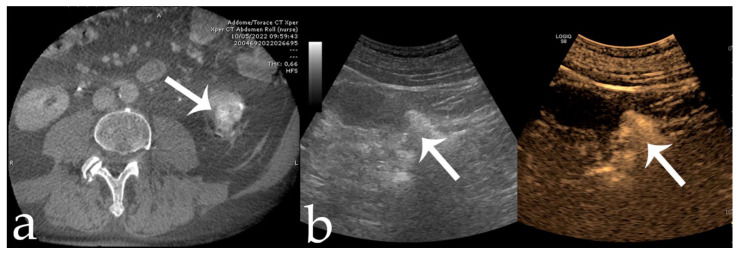
Same case as Figure 3. (**a**,**b**) Post-procedural CBCT and CEUS showing complete tumor devascularization (arrows).

**Figure 5 jcm-11-06816-f005:**
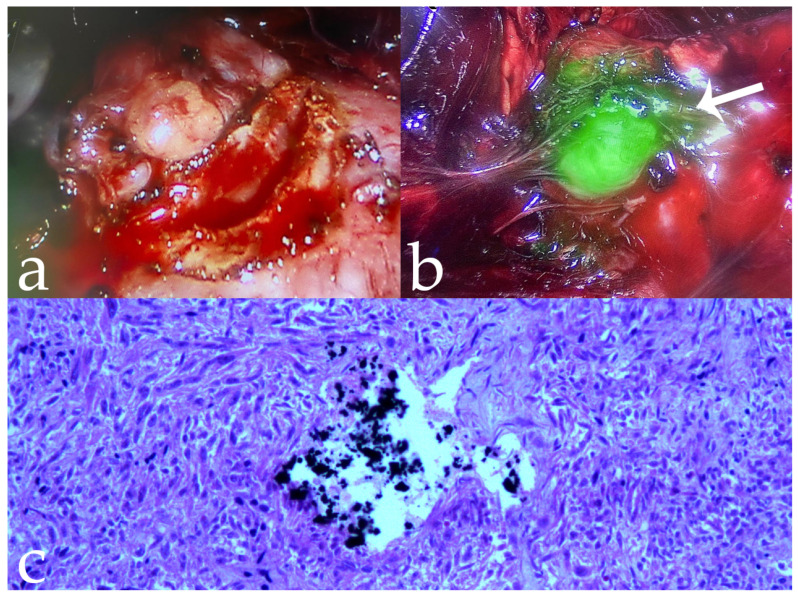
Same case as Figure 3 and Figure 4. (**a**,**b**) Tumor intra-operative visualization during robotic-assisted enucleation before and after fluorescence imaging (arrow); (**c**) histological findings showing deep penetration of 18-Onyx (10×).

**Figure 6 jcm-11-06816-f006:**
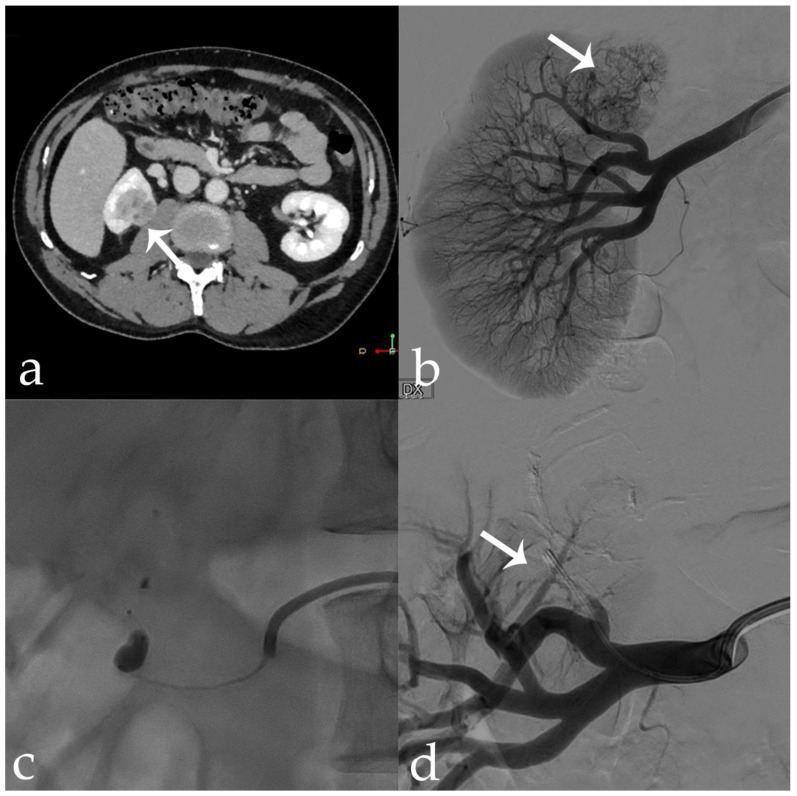
56 yr old male with endo-exophytic RCC of the right kidney upper pole (30 × 20 mm): (**a**) pre-operative CECT; (**b**) pre-embolization DSA showing tumor vascular supply (arrow); (**c**) embolization procedure using stop-flow technique after super selective setting with a microcatheter-balloon inflated during 18-Onyx and indocyanine mixture administration; (**d**) post-procedural DSA showing complete tumor devascularization (arrow).

**Figure 7 jcm-11-06816-f007:**
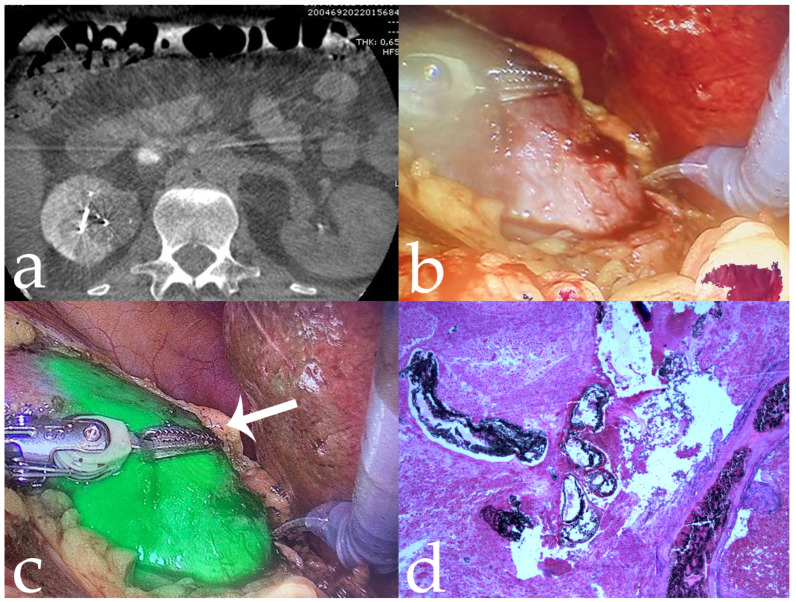
Same case as Figure 6. (**a**) Post-procedural CBCT showing complete tumour devascularization; (**b**,**c**) Tumor visualization before and after fluorescence imaging, during robotic-assisted partial nephrectomy (arrow); (**d**) surgical specimen with intralesional distribution of mixture embolic agent (10×).

**Table 1 jcm-11-06816-t001:** The main characteristic of the population and measured size of the pathological specimen.

	Value
Age (years)	72 (54–82)
Sex (F/M)	2/11 (15%/85%)
Histology	
RCC Oncocytoma Sarcomatoid RCC	9/13 (69%) 3/13 (23%) 1/13 (8%).
Size + SD (range)	29 mm ± 11.1 (15–50 mm)
Location	
Upper Middle third Lower	3/13 (23%) 3/13 (23%) 7/13 (54%).
Location	
Exophytic Endophytic	9/13 (69%) 4/13 (31%)
Mean blood loss + SD (range)	145 cc ± 101 (10–300 cc)
Preoperative serum creatinine (SD)	0.98 mg/dL (±0.3)
Postoperative serum creatinine (SD)	1.07 mg/dL (±0.3)

## Data Availability

The data presented in this study are available on request.

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
