# Peer review of "Green Tattoo Pre-Operative Renal Embolization for Robotic-Assisted and Laparoscopic Partial Nephrectomy: A Practical Proof of a New Technique"

_jcm, 2022, doi:10.3390/jcm11226816_

Round 1

Reviewer 1 Report

Reviewer: 

In the manuscript entitled “Green tattoo pre-operative renal embolization for kidney cancer robotic-assisted and laparoscopic partial nephrectomy with fluorescence imaging: a practical proof of a new technique”. In the manuscript authors have investigated that, Green tattoo pre-operative renal embolization for kidney cancer robotic-assisted and laparoscopic partial nephrectomy with fluorescence imaging. To prove their hypothesis authors have presented experimental data. Overall, I found the article interesting to read and the data presented will add value to the scientific community. I have some comments that need to answer.

Comments

1.     Title should be revised, “nefrectomy spelling is wrong

2.     The authors include histopathologic images (H&E staining) in Fig. 1i and 2j. that are so small and poor quality they are impossible to evaluate. 

3.     Would it be possible to quantify the intralesional penetration capacity from Fig. 1i and 2j.

4.     How author measure the Median tumor size.

5.     There are some format errors, which should be corrected, such as leave a space between number and unit.

6.     The authors should add a critical debate on the limitations of the present study to the Discussion.

7.     I will suggest adding a few more sentences in the conclusion section.

8.     There are many typo error in the text. The language needs to be polished.

Author Response

Reviewer: 

In the manuscript entitled “Green tattoo pre-operative renal embolization for kidney cancer robotic-assisted and laparoscopic partial nephrectomy with fluorescence imaging: a practical proof of a new technique”. In the manuscript authors have investigated that, Green tattoo pre-operative renal embolization for kidney cancer robotic-assisted and laparoscopic partial nephrectomy with fluorescence imaging. To prove their hypothesis authors have presented experimental data. Overall, I found the article interesting to read and the data presented will add value to the scientific community. I have some comments that need to answer.

Comments

  1. Title should be revised, “nefrectomyspelling is wrong. Thanks to the Reviewer for the suggestion. The word has been corrected through all the manuscript.
  2. The authors include histopathologic images (H&E staining) in Fig. 1i and 2j. that are so small and poor quality they are impossible to evaluate. Images have been revised and modified as suggested by the Reviewer.
  3. Would it be possible to quantify the intralesional penetration capacity from Fig. 1i and 2j. Thanks to the Reviewer for the comment. We specified in the discussion the following sentence: “The high intra-lesional penetration is proved by the presence of the ICG-Onyx-18 mixture in the renal tumor vascular supply; the entity of the embolization performance is strictly connected with the dimension and the number of the renal tumor supply branches mapped on the pathology specimen as showed in Figures 2d, 3e, 5c and 7d.”
  4. How author measure the Median tumor size. Thanks to the Reviewer for this comment. The tumor size was measured on the pathological specimen. We specified it in the M&M section.
  5. There are some format errors, which should be corrected, such as leave a space between number and unit. Thanks for the suggestion. The errors have been corrected.
  6. The authors should add a critical debate on the limitations of the present study to the Discussion. We described the limitations of the study on the last paragraph of the discussion section.
  7. I will suggest adding a few more sentences in the conclusion section. Thanks for this suggestion. The following sentence has been added in the conclusion section:” The technique is based on the deep penetration ability of a liquid embolic agent carrying ICG into the tumor vascular supply, resulting in excellent lesion devascularization and intra-operative tumor visualization and mapping with fluorescence imaging.”
  8. There are many typo error in the text. The language needs to be polished. Thanks for this comment. All the text underwent a revision.

Reviewer 2 Report

I congratulate the authors for the work done and to consider publishing “Green tattoo pre-operative renal embolization for kidney cancer robotic-assisted and laparoscopic partial nefrectomy with fluorescence imaging: a practical proof of a new technique”

General comments

-Suggest to shorten title 

- Also did you have a previous study with animal to evaluate safety and efficacy of the procedure before testing human trial.

- Solution of DMSO+Indocyanine+EVOH needs to be explained step by step in order to others use the same technique 

specific comments

Page1 line 41

Serum

Line 43

Indocyanine

Change in the entire manuscript

Page 2 line 53 - CKD - indicate the meaning of abbreviation.

Page2- line 100

CECT - please add contrast enhanced computer tomography

Page 3 Line 117-120

Paragraph is very confusing and is the most important part in the study. Suggest rewrite the entire paragraph step by step.

Page 3 Line 117

Do no start sentence with numbers

Page 4 Line 127

Change “In” to  “

Page 4 line 127

Microballoon

Page 4 Line 135

Balloon

Page 4 Line 143

Cone beam CT - please add

Page 4 Line 143

Contrast enhanced ultrasound

Line 153

Serum

Line 169

Serum

Figure 1

Very beautiful images. Suggest to arrange into 4x4 images with the same height and width. Eliminate D or E.

Figure 2

Also very beautiful images.

A. Name of the patient is visible please. Follow instructions to upload image i suggest 4x4 as maximum.

Figure 3.

Suggest the same as previous figures 4x4. Also just to include 1 or 2 cases and show different aspects or the important aspect to focus.

Or to change all figures and separate CT findings, Angiography findings and embolization procedure, CT or cone beam ct + CEUS and Surgical findings with hystology.

Also please cite in the main text the figure -1,2 and 3.

Line 276

Balloon

Author Response

Review 2

I congratulate the authors for the work done and to consider publishing “Green tattoo pre-operative renal embolization for kidney cancer robotic-assisted and laparoscopic partial nefrectomy with fluorescence imaging: a practical proof of a new technique”

General comments

-Suggest to shorten title. Thanks to the Reviewer for this suggestion. The title has been modified as suggested: “Green tattoo pre-operative renal embolization for robotic-assisted and laparoscopic partial nephrectomy: a practical proof of a new technique”

- Also did you have a previous study with animal to evaluate safety and efficacy of the procedure before testing human trial. No previous studies on animals have been performed. However, in literature there are already similar studies which proved the safety and efficacy of the technique.

- Solution of DMSO+Indocyanine+EVOH needs to be explained step by step in order to others use the same technique. Following Reviewer’s comment, the technique has been described step by step in the first paragraph of the section 2.2.

specific comments

Page1 line 41

Serum Thanks for the suggestion. The errors have been corrected.

Line 43

Indocyanine

Change in the entire manuscript Thanks for the suggestion. The word has been changed.

Page 2 line 53 - CKD - indicate the meaning of abbreviation. Thanks for the suggestion. The meaning of the abbreviation has been added.

Page2- line 100

CECT - please add contrast enhanced computer tomography Thanks for the suggestion. The meaning of the abbreviation has been added.

Page 3 Line 117-120

Paragraph is very confusing and is the most important part in the study. Suggest rewrite the entire paragraph step by step. The paragraph has been modified, as following: “Thirteen patients, candidates to robotic-assisted and laparoscopic partial nephrectomy, received pre-operative mixed indocyanine-ethylene vinyl alcohol (EVOH) (Green-embo) embolization between June 2021 and August 2022. All the patients enrolled in the study presented these inclusion criteria : (a) pre-operative contrast-enhanced computer tomography (CECT) and/or contrast-enhanced MRI; (b) biopsy-proven renal mass; (c) laparoscopic or robot-assisted partial nephrectomy according to current guidelines.”

Page 3 Line 117

Do no start sentence with numbers Thanks for the comment. We changed the text as suggested.

Page 4 Line 127

Change “In” to  “ corrected

Page 4 line 127

Microballoon corrected

Page 4 Line 135

Balloon corrected

Page 4 Line 143

Cone beam CT - please add  corrected

Page 4 Line 143

Contrast enhanced ultrasound corrected

Line 153

Serum corrected

Line 169

Serum corrected

Figure 1

Very beautiful images. Suggest to arrange into 4x4 images with the same height and width. Eliminate D or E.

Thanks for this comment. The images has been modified following the Reviewer suggestion.

Figure 2

Also very beautiful images.

  1. Name of the patient is visible please. Follow instructions to upload image i suggest 4x4 as maximum. Thanks for this comment. The images has been modified following the Reviewer suggestion.

Figure 3.

Suggest the same as previous figures 4x4. Also just to include 1 or 2 cases and show different aspects or the important aspect to focus.

Or to change all figures and separate CT findings, Angiography findings and embolization procedure, CT or cone beam ct + CEUS and Surgical findings with histology.

Also please cite in the main text the figure -1,2 and 3.

Thanks for these comments. All the images have been modified following the Reviewer suggestions.

Line 276

Balloon Corrected.

Round 2

Reviewer 2 Report

Dear authors thank you for your quick improvement of the manuscript

Just to remark something important.

Embolization technique: it still remains uncomprehensive

"Before the embolization procedures, a mixture consisting of indocyanine (powder 25 mg, diagnostic green), and OnyxTM 18 LES (Medtronic) was prepared. The first step is to mix 0.7 ml of DMSO and add to the indocyanine tube. Then 0.5 ml of the resulting mixture (DMSO + Indocyanine) is then administered ...........

Here is my question. Typically the DMSO is injected inside the microcatheter using the exact volume for the dead space, and then the shaken Onyx is injected slowly into the vessels. Don't understand what it means "administered into the row containing OnyxTM 18 LES and subjected to automatic mixing for about 20 minutes before the procedure. 

Did you mix the DMSO and Onyx together and create a new mixture of 1cc OnyxTM 18 + 0.5 (DMSO+Indocyanine).  How the embolization procedure was performed exactly. Is the most important part of the manuscript to explain in detail how this mixture and embolization procedure was performed. 

Author Response

Dear Reviewer. Thanks for your suggestion. The text has been modified as follows:

“Before the embolization procedure, a mixture consisting of indocyanine (powder 25 mg, diagnostic green) and OnyxTM 18 LES (Medtronic) is prepared. The operator has one vial of DMSO, one vial of Onyx-18 and one of indocyanine in powder available. The first step is to septically withdraw 0.7 ml from the DMSO vial and inject them into the container of indocyanine powder, mixing. Then 0.5 ml of the new compound consisting of DSMO and indocyanine are taken and administered in the vial containing OnyxTM 18. The new mixture obtained, therefore consisting of 0.5 ml of DSMO-indocyanine and a vial of OnyxTM-18, after being mixed automatically for about 20 minutes, will be used to carry out the pre-operative embolization. The residual volume of the DSMO and possibly another vial of DSMO, if necessary, will be used to fill the dead volume of the microcatheter, before the administration of the embolizing mixture.”
